# From Cure to Prevention: Doxycycline’s Potential in Prophylaxis for Sexually Transmitted Infections

**DOI:** 10.3390/antibiotics13121183

**Published:** 2024-12-05

**Authors:** James Bird, Basil Alawyia, Nikolaos Spernovasilis, Danny Alon-Ellenbogen

**Affiliations:** 1Department of Basic and Clinical Sciences, University of Nicosia, Nicosia 2417, Cyprus; alawyia.bas@live.unic.ac.cy (B.A.); alon-ellenbogen.d@unic.ac.cy (D.A.-E.); 2Department of Infectious Diseases, German Medical Institute, Limassol 4108, Cyprus

**Keywords:** doxycycline, sexually transmitted infections, prophylaxis

## Abstract

Over the past two decades, the global incidence of sexually transmitted infections (STIs) such as gonorrhea, chlamydia, and syphilis have increased significantly, particularly among cisgender men who have sex with men (MSM) and transgender women (TGW). This rise in STIs has spurred interest in new preventive measures, including doxycycline post-exposure prophylaxis (DoxyPEP). Clinical trials in the United States and France have demonstrated the effectiveness of DoxyPEP in reducing both chlamydia and syphilis incidence among MSM and TGW; although, its efficacy against gonorrhea remains limited, and it was further found to be ineffective among cisgender women in Kenya. Due to the promising results, the CDC and the German STI Society have incorporated DoxyPEP into their guidelines for specific high-risk groups. However, the broader implementation of DoxyPEP presents several challenges and ethical concerns. Key issues involve the potential development of antimicrobial resistance, particularly among common STI pathogens like *C. trachomatis*, *M. genitalium*, and *N. gonorrhoeae*, as well as other bacteria such as *S. aureus* and *K. pneumoniae*. Additionally, questions concerning equitable healthcare access, the potential impact on adherence to safer sex practices, and broader public health implications warrant careful consideration. Addressing these challenges necessitates a careful balance between the benefits and risks of DoxyPEP, as well as the implementation of strategies to mitigate negative outcomes while maximizing the impact on public health. Lastly, future research should explore the integration of DoxyPEP with other preventive strategies, such as vaccines, to enhance its effectiveness and reduce the global burden of STIs.

## 1. Introduction

Over the past two decades, sexually transmitted infections (STIs) such as *Neisseria gonorrhoeae* (gonorrhea), *Chlamydia trachomatis* (chlamydia), and *Treponema pallidum* (syphilis) have been gradually increasing globally. This upward trend in STI incidence has been attributed to various factors, including the increase in the number of sexual partners and risky sexual behavior with reduced condom use; furthermore, the introduction of human immunodeficiency virus (HIV) pre-exposure prophylaxis (PrEP) is thought to have contributed to these increases [1]. Nonetheless, HIV PrEP has demonstrated effectiveness in protecting against HIV and has shown that biomedical interventions for STI prevention can be safe, effective, and accepted by the community [1].

Moreover, the “test and treat” model—implemented in most high-income countries to combat this rise in STIs—has not been successful [2]. This lack of success has shifted attention towards a proactive public health strategy focused on prevention rather than treatment. The idea of antibiotic prophylaxis against STIs has been implemented before in low- and middle-income countries in the form of periodic presumptive treatment, which was given to high-risk individuals such as female sex workers or young heterosexuals. These antibiotic regimens often consist of azithromycin with or without ciprofloxacin and cefixime. While, theoretically, this strategy should have been effective, several factors, particularly antimicrobial resistance (AMR), resulted in only a modest reduction in the incidence of gonorrhea and chlamydia but not syphilis [3]. Furthermore, in the 1940s, the idea of antibiotic prophylaxis against STIs was proposed to reduce the incidence of gonorrhea in the US Navy; however, this only yielded transient changes and resulted in rapid selection and dissemination of antibiotic resistance to sulfonamides, minocycline, and penicillin [4]. Despite the success of 4CMenB vaccine against *N. meningitides*, showing 71–95% vaccine effectiveness in infants [5], a similar success has not been seen in *N. gonorrhea*. Additionally, there no current vaccines available for chlamydia and syphilis, among other STIs [4].

Given the lack of effective vaccines and the unsuccessful “test and treat” strategy combined with the success of HIV PrEP, the concept of STI antibiotic prophylaxis has been gaining momentum. Doxycycline is regarded as the antibiotic of choice, due to its broad coverage against a variety of STI-causing bacteria along with its pharmacokinetic properties and its tolerability. There are two suggested regimens of prophylactic dosing: pre- and post-exposure prophylaxis. This review aims to discuss several aspects of doxycycline as post-exposure prophylaxis, including the clinical evidence supporting its use, the challenges associated with widespread implementation of this strategy, as well as real-world case studies and future developments.

## 2. Scope of the Problem

STIs remain a significant public health challenge, with a concerning trend of increasing incidence rates across the globe. According to the World Health Organization (WHO), in 2020, STIs such as gonorrhea, chlamydia, and syphilis accounted for more than 374 million cases worldwide [6], underscoring their widespread impact on global health.

The US Centers for Disease Control and Prevention (CDC) have observed a consistent year-on-year increase in the rates of these STIs. However, there was a notable exception from 2020 to 2021, in which the implementation of lockdowns and social distancing measures due to the COVID-19 pandemic led to a temporary decrease in STI prevalence rates [7]. This pattern was similarly observed by the European Centre for Disease Prevention and Control (ECDC) [8,9,10], indicating a global disruption in STI screening during the pandemic [11].

The WHO has highlighted specific regions as areas of particular concern due to their high prevalence rates of STIs [6]. For example, countries in Africa have some of the highest global rates of syphilis and gonorrhea. In contrast, the Americas—encompassing both North and South America—show the highest global prevalence rates of chlamydia. Notably, the global prevalence of chlamydia far exceeds that of gonorrhea and syphilis, making it the most widespread bacterial STI [6].

The disparities in the prevalence of these diseases between genders further complicate the global STI landscape. Females are disproportionately affected by gonorrhea and chlamydia, while the prevalence of syphilis is more evenly distributed between males and females [6]. This gender disparity is concerning, as women are more likely to experience severe complications from STIs, including pelvic inflammatory disease, infertility, and ectopic pregnancies [12,13]. Furthermore, the trend in increasing STI rates is seen to be highest in men who have sex with men (MSM) and transgender women (TGW). These disparities underscore the need for targeted prevention strategies and healthcare interventions to address the unique needs of different populations and to mitigate the long-term health consequences of STIs.

STIs significantly increase the risk of acquiring HIV. The presence of an STI can make individuals more susceptible to HIV infection by compromising the integrity of mucosal barriers and increasing the concentration of HIV target cells in the genital tract [14]. This link between HIV and STIs further highlights the importance of addressing the STI epidemic as part of a broader public health strategy to combat global HIV/AIDS.

## 3. DoxyPEP Concept Explained

Doxycycline is a second-generation tetracycline that inhibits the ribosomal synthesis unit, resulting in a bacteriostatic effect. When administered orally, it is almost completely absorbed in the gastrointestinal tract. Moreover, its absorption is not significantly affected by the presence of milk or food and results in a half-life of 20 h, allowing once- or twice-daily dosing. It is distributed throughout the body, including the liver, kidneys, digestive tract, and semen, and is eliminated through non-renal mechanisms, making it safe in patients with renal failure [15].

According to CDC guidelines, doxycycline is the first-line treatment for *M. genitalium* infections, along with moxifloxacin or azithromycin, depending on the sensitivity, and for *C. trachomatis* infections [16]. However, it is no longer recommended for treating *N. gonorrhoeae* infections due to high resistance rates, particularly in Europe [15]. With regard to syphilis treatment, doxycycline is a second-line treatment, often used in cases of penicillin allergy or supply shortages [17].

Doxycycline has been successfully employed as a post-exposure prophylaxis against borreliosis and leptospirosis [18,19] and as a pre-exposure prophylaxis against malaria, particularly among those traveling to endemic areas [20]. Therefore, based on its medical uses and pharmacokinetic profile, doxycycline has emerged as a suitable antibiotic for prophylaxis against STIs and is referred to as DoxyPEP, which is defined as 200 mg of oral doxycycline taken within 72 h of condomless sex [21].

## 4. Side Effects

Doxycycline is generally well tolerated, with most patients experiencing only mild side effects [22]. The most frequently reported adverse effects involve the gastrointestinal system, affecting up to 55% of patients. These symptoms can include diarrhea, vomiting, nausea, and pill-esophagitis. Skin-related side effects are also noted, with photosensitivity occurring in fewer than 47% of cases, while Stevens–Johnson syndrome has been reported in less than 1% of these [23]. Other potential side effects include hematologic toxicity such as hemolytic anemia, thrombocytopenia, eosinophilia, and neutropenia. However, these side effects are rare and more frequently associated with other tetracyclines [22]. Importantly, the most severe adverse effects associated with doxycycline are typically reversible upon discontinuation of the drug, with the severity of side effects being related to the duration of treatment and dosage [15].

Tetracyclines are contraindicated during pregnancy due to the risk of teratogenic effects and impaired bone growth, caused by calcium chelation, although there has been some debate about whether these risks apply equally to all tetracyclines, as some studies have indicated that therapeutic doses of doxycycline are unlikely to be teratogenic [24,25]. Additionally, any impairment in bone growth has been shown to normalize after discontinuation of the drug [26]. Despite this, the safety of doxycycline during pregnancy has not been conclusively established, and the British National Formulary continues to advise against the use of all tetracyclines during pregnancy whenever possible [27]. Clinical trials investigating the use of doxycycline as DoxyPEP typically recorded low drop-out rates [18,28,29] and, in the case of Stewert et al. (2023), mostly due to pregnancy. Furthermore, in all three trials, no serious adverse events were reported [18,28,29].

## 5. Studies Assessing the Efficacy of DoxyPEP

Several clinical trials have assessed the efficacy, safety, and risk of resistance associated with DoxyPEP (Table 1). A substudy of the French ANRS-IPERGAY trial conducted in 2015 assessed the efficacy of DoxyPEP in HIV-negative MSM and TGW who engage in sex with men [18]. Patients were given 100mg of oral doxycycline to be used within 72 h of high-risk sexual intercourse, such as condomless or oral sex. The occurrence of a first STI was 47% lower in patients taking PEP than in those not taking PEP (HR: 0.53 [95% CI 0.33–0.85]; *p* = 0.008). More specifically, patients taking PEP had a lower incidence of chlamydia (HR: 0.30 [95% CI 0.13–0.70]; *p* = 0.006) and syphilis (HR: 0.27 [95% CI 0.07–0.98]; *p* = 0.047) than those not taking PEP. On the other hand, the incidence of gonorrhea did not differ significantly between both groups (HR: 0.83 [95% CI 0.47–1.47]; *p* = 0.52) (Table 1). This lack of difference can be partially explained by the high levels of tetracycline-resistant gonorrhea in France, which may reduce the efficacy of DoxyPEP.

Furthermore, a study conducted by the National Institutes of Health in the US assessed the efficacy of DoxyPEP in MSM and TGW taking HIV PrEP and patients living with HIV (PLWH) [30]. The study spanned from August 2020 to May 2022. It revealed a reduction in the incidence of STIs in both the HIV PrEP (RR: 0.34 [95% CI 0.24–0.46]; *p* < 0.001) and the PLWH (RR: 0.38 [95% CI 0.24–0.60]; *p* < 0.001) groups that were taking DoxyPEP (Table 1). In contrast to the ANRS-IPERGAY trial, the incidence of gonorrhea was reduced in both the HIV PrEP (RR: 0.45 [95% CI 0.32–0.65]) and PLWH (RR: 0.43 [95% CI 0.26–0.71]) groups that were taking DoxyPEP. This result reflects the difference in the levels of tetracycline-resistant gonorrhea between France and the US. Similarly, the incidence of chlamydia and syphilis was reduced in both groups that were taking DoxyPEP. Interestingly, the French ANRS DOXYVAC study conducted in 2022 [28] assessed the efficacy of DoxyPEP in MSM on HIV PrEP and revealed that DoxyPEP was associated with a significant reduction in the incidence of chlamydia (aHR: 0.13 [95% CI 0.08–0.21]; *p* < 0.0001) and syphilis (aHR: 0.22 [95% CI 0.11–0.43]; *p* < 0.0001). On the other hand, the study showed moderate, but lower, efficacy against gonorrhea (aHR: 0.63 [95% CI 0.50–0.79]; *p* < 0001) (Table 1) when compared to the US DoxyPEP study [30]. This further reflects the effect of tetracycline resistance on the efficacy of DoxyPEP against gonorrhea.

The studies mentioned thus far have not included cisgender women, with MSM and TGW being the primary focus, reflecting the population of patients that is most notably affected by STIs. Nonetheless, an open-label randomized trial in Kenya conducted between February 2020 and October 2022 focused on the efficacy of DoxyPEP in cisgender women receiving HIV PrEP [29]. In contrast to the previous studies, this study revealed no significant difference in the incidence of STIs between cisgender women who took DoxyPEP and those who did not (RR: 0.88 [95% CI 0.60–1.29]; *p* = 0.51). Several factors have contributed to this result, mainly low adherence and high prevalence of tetracycline-resistant *N. gonorrhoeae*, as well as sexual network dynamics including frequency of sexual exposure and lack of services for testing and treating partners. Moreover, it has been proposed that differences in primary infection site may contribute to the differences in DoxyPEP efficacy between cisgender women and MS< and TGW. In cisgender women, the primary site of infection is the endocervix, while in MSM and TGW, the primary sites of infection are rectal tissue and pharyngeal tissue. However, recent evidence has suggested that doxycycline concentration in the vaginal tissue was sufficient to prevent STIs. This finding suggests that a factor other than differences in the site of infection may contribute to the difference in efficacy; nonetheless, this would need further investigations [29].

A meta-analysis by Szondy et al. assessed the current state of evidence on the efficacy of DoxyPEP. Among other studies, the meta-analysis included the ANRS-IPERGAY trail [18], the US DoxyPEP trial [30], the DoxyPEP Kenya Trial [29], and the DOXYVAC trial [28]. The results of the analysis remained consistent with the findings in each of the studies, with an increased incidence of STIs in the control group (RR: 0.47 [95% CI 0.29–0.77]). More specifically, the incidences of chlamydia (RR: 0.26 [95% CI 0.10–0.68]), syphilis (RR: 0.23 [95% CI 0.14–0.36]), and gonorrhea (RR: 0.66 [95% CI 0.34–1.26]) were all lower in the DoxyPEP group when compared to the control group [31].

## 6. Antibiotic Resistance

One of the main challenges facing the widespread implementation of DoxyPEP is the emergence of AMR. Despite its efficacy, DoxyPEP could potentially lead to increased rates of doxycycline-resistant STIs [32]. Bacteria acquire resistance to tetracyclines through several mechanisms. These include decreased entry via mutations in porin proteins, increased activity of efflux pumps, and mutations at specific sites of the 30S ribosomal subunit and 16S rRNA [33]. With doxycycline-resistant gonorrhea already prevalent, DoxyPEP can further exacerbate this issue and increase the risk of cross-resistance in gonorrhea. Additionally, given that doxycycline is a common and effective treatment option against chlamydia and *M. genitalium*, the development of tetracycline resistance might prove a serious public health threat, as this would significantly reduce the available treatment options. Nevertheless, currently, there have been no documented clinical cases of doxycycline-resistant chlamydia, syphilis or *M. genitalium*.

Few studies have assessed the risk of increased rates of AMR among STIs in the context of DoxyPEP. The ANRS DOXYVAC study found a significant difference in rates of high-level tetracycline-resistant *N. gonorrhoeae* isolates at the end of the study (35.5% DoxyPEP vs. 12.5% control, *p* = 0.043). In all isolates obtained during the study, some level of tetracycline resistance was identified; however, it is important to note that only 78 isolates were available, serving as one of the limitations of this study [28]. In the case of *N. gonorrhoeae*, the San Francisco DoxyPEP clinical trial found tetracycline resistance increased from 2/7 (28.6%) patients at baseline to 5/13 (38.5%) at the end of the study in the DoxyPEP group, whereas resistance rates did not increase in the control group (2/8 (25%) at baseline, 2/16 (12.5%) at the end of the study) [30]. All *N. gonorrhoeae* isolates in Kenya were found to be resistant to tetracycline [29]. For *C. trachomatis*, both the US DoxyPEP trial and the ANRS-IPERGAY study found no evidence of tetracycline resistance [18,30]. Mutations in *M. genitalium* that could confer resistance to tetracyclines were identified in the DoxyPEP group in the ANRS-IPERGAY study [18]. This was based on in vivo analysis and was not associated with treatment failure; hence, further investigations would be necessary. Due to the inherent challenges in culturing *T. pallidum*, resistance has not yet been assessed [34]. Nonetheless, in vitro studies have shown that doxycycline exhibits anti-treponemal activty (MIC 0.10 mg/L) [17]. In summary, currently, only *N. gonorrhea* has developed tetracycline resistance, which may be further exacerbated by the use of DoxyPEP. Nonetheless, further studies with longer follow-ups are needed to accurately determine the risk of AMR with enrollment of DoxyPEP.

Furthermore, as DoxyPEP is associated with increased antibiotic exposure, there is an increased risk of developing tetracycline resistance among commensal bacteria. The US trial initially isolated *Staphylococcus aureus* from 45% of participants, of which 12% of isolates were found to be resistant to doxycycline. At the end of the study, *S. aureus* was only isolated in 28% of individuals, of which 16% were resistant. In the control group, *S. aureus* was isolated in 47% of individuals, of which 8% were resistant [30]. Moreover, an in vitro passage study in a *Galleria mellonella* model of doxycycline PEP revealed that tetracycline resistance in *Escherichia coli* and *Klebsiella pneumoniae* emerged rapidly [35]. Furthermore, unlike other tetracyclines, *Streptococcus pneumoniae* is not resistant to doxycycline; however, the mechanism by which *S. pneumoniae* acquired resistance against other tetracyclines is similar to how *N. gonorrheae* acquired doxycycline resistance. As a result, there is a hypothetical risk that with increased doxycycline exposure, *S. pneumoniae* can acquire doxycycline resistance. This is particularly important as there has also been a rise in penicillin-resistant *S. pneumoniae* [36].

In addition to promoting resistance for tetracycline, DoxyPEP could further promote resistance to other antibiotic classes, such as fluoroquinolones and penicillin, among others. An analysis of minimum inhibitory concentration (MIC) data from the Antimicrobial Testing Leadership and Surveillance (ATLAS) database assessed the correlation between minocycline MICs and MICs of other antibiotics. Minocycline, which closely resembles doxycycline, was found to have MICs with a positive association to those of ceftriaxone, ampicillin, levofloxacine and amikacin for the majority of Gram-negative and Gram-positive bacteria in the database. This analysis suggests that, at least theoretically, DoxyPEP could result in the emergence of antimicrobial resistance to these classes of antibiotics [37]. This is particularly important for *N. gonorrhoeae* as ceftriaxone is the recommended first-line treatment.

An analysis of whole genome sequencing data of 2375 gonococcal isolates from the 2018 Euro-GASP survey revealed that tetracycline-resistant *N. gonorrhoeae* contained mutations that could select for resistance against other antimicrobials, including ceftriaxone. Moreover, this analysis showed that strains of tetracycline-resistant *N. gonorrhoeae* often exhibited resistance or reduced susceptibility to ceftriaxone. Based on that, DoxyPEP could potentially select for ceftriaxone resistance; however, this would depend on other factors including how widespread DoxyPEP use is, the population of patients using both, and the extent of *N. gonorrhoeae* AMR. Nonetheless, no clinical evidence of this has been reported [38].

As previously mentioned, there have been no reported clinical cases of doxycycline-resistant chlamydia. However, a study investigated *C. suis* isolated from pigs for tetracycline-resistant genes, and found that tet(C) was present in all isolates. The exact mechanism in which these isolates acquired resistance is unknown. This finding reflects the theoretical possibility that *C. trachomatis* could acquire tetracycline resistance [39]. Unlike other bacteria, *T. pallidum* does not acquire resistance through gene transfer, but rather via point mutations. This has already been observed within the 23s rRNA gene resulting in azithromycin resistance. In other organisms, a similar mutation in the 16s rRNA gene results in doxycycline resistance. Moreover, studies have shown that odds of *T. pallidum* resistance were higher in individuals who had taken macrolide therapy in the preceding year. Accordingly, concerns still remain about the emergence of such mutations with the widespread use of DoxyPEP [40].

## 7. The Concept of “Arrested Immunity”

Another consideration associated with implementing DoxyPEP is the potential risk of disrupting the body’s immune response against infections, thereby preventing the development of long-term immunity against STIs and subsequently increasing the risk of reinfection. This hypothetical concept is known as the “Arrested Immunity Hypothesis” and was initially based on the interaction between the immune system and chlamydia [41]. An immunoepidemiologic model was developed by Vickers and Osgood, in 2014, to explore this interaction to better understand this hypothesis [42]. The results of this model showed that early treatment does impair the development of an immune response. However, other factors such as the sexual practices of the infected individual, the number of people treated, and the timing of the treatment can alter the burden of infection and alter the host’s immune response [42]. In line with this, DoxyPEP may theoretically reduce immunity to chlamydia, leading to increased reinfection rates; however, all studies assessing the efficacy of DoxyPEP showed a reduction in the incidence of STIs, including reinfections [34]. Ultimately, more studies are needed to determine if the Arrested Immunity Hypothesis is implicated in the use of DoxyPEP.

## 8. Current Guidelines

Globally, there are several different guidelines, with varying recommendations as to the use of DoxyPEP. The guidance from the British Association for Sexual Health and HIV, last updated in 2021, advises against routinely prescribing doxycycline as prophylaxis for STIs [43]. This recommendation was based on several considerations that were available at the time, including the lack of conclusive evidence supporting the effectiveness of doxycycline for STI prevention and concerns about the potential for AMR. Tetracycline resistance has already been observed in *T. pallidum*, and the high rates of resistance in *N. gonorrhoeae* further reduce the potential benefits of DoxyPEP. Additionally, the risk of *S. aureus* developing resistance due to widespread doxycycline use was also a significant concern [43]. Other considerations included the lack of patient supervision when prescribing DoxyPEP, which could lead to issues in the case of pregnancy or rare adverse effects [43].

The San Franscisco Department of public health was the first to recommend the use of DoxyPEP for the prevention of STIs. Their guidelines suggest offering prophylaxis to MSM and TGW who engage in condomless anal or oral sex [44]. However, they advised against routine use of doxycycline for individuals who engage in receptive vaginal sex due to insufficient evidence. The recommended dosage is 200 mg of doxycycline, ideally taken within 24 h, but no later than 72 h after sexual contact. They also advise limiting the dose to 200 mg within a 24 h period, with follow-up every three months for STI screening as well as liver and renal function tests while using DoxyPEP.

The New York City department of Health and Mental Hygiene updated their guidelines in 2023 to recommend DoxyPEP (200 mg) for MSM and TGW, who have a history of a bacterial STI within the previous year. This should be administered within 24–72 h of condomless sex and should not be routinely offered to transgender men and to cisgender women [45].

The Australasian Society for HIV Medicine updated their recommendations in 2023. The recommendations are that DoxyPEP is suitable for MSM individuals with a recent history of syphilis, or with a recent history of two or more STIs that are not syphilis. Furthermore, MSM with concurrent male and female sexual partners or other sexual partners with a uterus are to be offered DoxyPEP. Additionally, MSM with an upcoming period of high-risk sexual intercourse, such as a sex event, or holiday plans that involve multiple sexual partners, may be offered DoxyPEP. This guidance recommends that DoxyPEP primarily be recommended for the prevention of syphilis and, to a lesser degree, for the prevention of chlamydia and gonorrhea. It is further recommended to use DoxyPEP for a pre-defined period of time, such as three or six months [46].

The German STI Society has released a statement regarding the use of DoxyPEP, according to which individuals must fulfill two requirements: DoxyPEP should be offered only to MSM or to TGW who have sex with men and who are either using HIV PrEP or are living with HIV [47].

The most recently updated guidelines come from the CDC, which recommend that MSM, or TGW who have sex with men, with a history of at least one bacterial STI in the last 12 months should be counseled on the benefits and harms of DoxyPEP. If suitable, prophylaxis should be taken within 72 h of exposure, at a dose of 200 mg. The individual should further be advised not to exceed 200 mg per 24h period, and the ongoing need should be assessed every three to six months [48]. Additionally, the CDC emphasizes that doxycycline prophylaxis should be part of a comprehensive sexual health strategy, including regular STI screening, vaccination, HIV testing, and counseling. The recommendation excludes cisgender women, cisgender heterosexual men, transgender men, and other queer and non-binary individuals [48].

The European AIDS Clinical Society updated their guidelines in 2023 to recommend that DoxyPEP be considered in MSM with a history of STIs. The guidelines state that there is a caveat that there may be unknown long-term effects on microbiota and AMR in STIs [49].

The International Union Against Sexually Transmitted infections: Europe updated their position statement in regards to DoxyPEP in June 2024. The statement agrees that DoxyPEP offers a benefit at an individual level, and that recommendations for it should take into account several factors. Firstly, these include the local STI epidemiology and the potential impact DoxyPEP may have. Secondly, they include the ability for healthcare services to offer DoxyPEP with suitable follow-ups, while not impacting established provisions for STI screenings and care. Thirdly, health systems integrating DoxyPEP should have the capacity to monitor AMR in both STIs and other bacterial infections. Lastly, discussions with the users must be established, to ensure that decision-making involves the users, and that understanding of the potential benefits and harms of DoxyPEP is established [50]. These recommendations further state that individuals who may benefit from DoxyPEP may vary between jurisdictions, and that trial evidence is currently limited to MSM and TGW who have sex with men, and therefore, recommendations cannot be made for individuals outside of these populations.

## 9. Challenges to Widespread Implementation

As previously stated, San Francisco was the first city to recommend and implement the use of DoxyPEP for MSM and TGW in October 2022, based largely on the results of the US DoxyPEP clinical trial. The results of this widespread implementation were presented at the Conference on Retroviruses and Opportunistic Infections (CROI) 2024. According to data from the San Francisco AIDS Foundation’s Magnet Sexual Health clinic, DoxyPEP was offered to around 3000 HIV PrEP users in late November 2022 with 1209 individuals (39% of all HIV PrEP users) opting to use it by September 2023. The overall incidence rate of STIs among DoxyPEP users declined from 18% to 8%, representing a 58% decrease. In contrast, the incidence rate of STIs among individuals not on DoxyPEP fell from 8% to 7% [51].

Similar to the clinical trials, the decrease in incidence for chlamydia (67%) and syphilis (78%) was significant, unlike the decrease in gonorrhea cases (11%), which was not statistically significant [51]. Moreover, data from City Clinic (San Francisco’s main sexual health clinic) revealed a decline in the incidence of both chlamydia and syphilis in DoxyPEP users compared to non-users. Notably, the difference here was not statistically significant, but this could be attributed to the small number of cases. On the other hand, the decline in gonorrhea cases was smaller in the DoxyPEP group; however, the difference was not statistically significant. Based on these data, it can be concluded that the clinical efficacy of DoxyPEP observed in clinical trials can be replicated in real-world settings [51]. Moreover, the success of public health strategies relies not only on their clinical effectiveness, but also on the willingness and acceptance of the targeted population.

In Australia, an online survey targeting MSM assessed the potential acceptability of syphilis chemoprophylaxis [52]. Of the 2095 participants, 52.7% (95% CI 50.6–54.8%) expressed an interest in using antibiotic prophylaxis to reduce the risk of syphilis infection. Additionally, in Canada, a questionnaire was offered during routine STI clinic visits, and among the 424 MSM who completed it, 60.1% were likely to use DoxyPEP [52]. Similarly, in the US, an online survey that targeted MSM revealed that among the 212 participants, 67.5% indicated they would consider DoxyPEP if recommended by their provider [52].

In Germany, an online survey was conducted to determine the extent to which the MSM community was already using DoxyPEP [53]. The survey included 99 participants, of whom 22 indicated that they had used doxycycline prophylactically from leftover medication from previous treatments. In addition, some individuals acquired doxycycline for prophylactic use through websites that sell it with or without a prescription. Through this, it can be speculated that this practice takes place in other countries across the western hemisphere.

Given the real-world clinical effectiveness of DoxyPEP in San Francisco and the general acceptance among target populations, DoxyPEP appears to be a promising public health strategy. However, a few factors must be considered before it can be implemented on a larger scale. These include the following: how patient populations that have not been studied in clinical trials, such as cisgender men and women, should be approached; what are the optimal intervals for screening for STIs; what is the maximum number of DoxyPEP doses allowed per month; and whether existing surveillance systems are sufficient to detect the emergence of AMR given the potential increase in the load of doxycycline in the environment [54].

Furthermore, as DoxyPEP serves to promote sexual health, there are valid parallels to HIV PrEP to consider avoiding potential pitfalls. HIV PrEP was underutilized in minority populations across the US, particularly African Americans and Hispanics. This underutilization was linked to structural and economic barriers such as racism, lack of insurance, stigma, homophobia, medical mistrust, and provider biases [55]. As such, a large portion of the target population for HIV PrEP did not receive its benefits. On that basis, ensuring that minority groups within the target population receive DoxyPEP is another important consideration for widespread implementation. This can be achieved through large-scale community engagement with the leadership of minority groups. In addition, ensuring that clinicians and healthcare providers are culturally sensitive is another essential component.

On a global scale, similar issues arise in low- and middle-income countries. Widespread poverty and lack of education and awareness serve as additional barriers for implementing DoxyPEP. The authors of a qualitative study assessing DoxyPEP adherence in women in western Kenya presented its findings in the International AIDS society (IAS) conference 2024. The study revealed that the main barriers to DoxyPEP adherence were side effects; mainly nausea, misunderstanding dosage instructions, and fear of partner reaction discouraged proper adherence. On the other hand, the study revealed that the perceived value of preventing an STI, familiarity with doxycycline, and the use of a discrete pill case motivated some women to adhere to DoxyPEP. These findings highlight the complexity of implementing DoxyPEP in some parts of the world, emphasizing the importance of ensuring patient understanding, being culturally competent, and battling stigma [56]. Moreover, due to the lack of STI screening protocols and testing centers, particularly for asymptomatic patients, implementing DoxyPEP would require additional measures to identify infected patients. Furthermore, the effects of intermittent antibiotic exposure in this subgroup of patients need to be evaluated, specifically concerning AMR. Lastly, to overcome these challenges, further controlled implementation of DoxyPEP, similar to what was carried out in San Francisco, is needed, with a longer follow-up period [55].

## 10. Ethical Considerations

The individual benefits of DoxyPEP are well documented, with multiple trials demonstrating its effectiveness in reducing the incidence of STIs [18,28,30]. Despite these promising results, it is crucial to consider the ethical implications and broader health concerns when evaluating the potential widespread use of DoxyPEP.

Current research has primarily focused on high-risk individuals, such as MSM with a prior history of STIs and a high number of sexual partners. For example, the ANRS-IPERGAY and ANRS DOXYVAC studies involved participants who had an average of 10 partners in the past three months and two months, respectively [18,28]. These figures are notably higher than the average number of sexual partners typically reported in the general MSM population [57,58,59]. This raises ethical questions regarding the applicability of DoxyPEP and its use in other populations. In populations with a lower STI burden, the number needed to treat would likely increase, potentially diminishing the overall public health benefit and raising concerns about the cost-effectiveness and broader applicability of DoxyPEP [30].

STI screening is a vital part of preventative healthcare, which is highlighted in many of the guidelines recommending DoxyPEP. The Belgium GONOSCREEN study compared the effect of screening for gonorrhea and chlamydia to non-screening [60]. The study demonstrated that non-screening was associated with a higher incidence of asymptomatic chlamydia infections, emphasizing the role of regular screening in identifying and treating these cases before they contribute to ongoing transmission. However, it also highlighted that screening did not significantly reduce gonorrhea incidence. Notably, this study was associated with increased antimicrobial consumption, raising ethical concerns regarding the overprescription of antibiotics, mirroring reservations surrounding the wider use of DoxyPEP [60].

Follow-up care is crucial for individuals who use DoxyPEP. Follow-up appointments are typically scheduled every 3 to 6 months for STI screening and general health evaluations. The CDC emphasizes the importance of utilizing DoxyPEP within a “comprehensive sexual health approach”, which incorporates counseling on the benefits and harms of doxycycline, as well as its limitations. Furthermore, counseling should include education regarding correct condom use and general risk reduction strategies. The increasing global incidence of STIs, which has risen following an initial decline during the early years of the HIV/AIDS epidemic, correlates with reduced condom use. Studies have shown that individuals are progressively less likely to use condoms, contributing to the rising STI rates [61,62].

Another significant concern is the potential teratogenic effect of doxycycline. Although evidence of teratogenicity comes from animal studies, there is a lack of human data on this topic. If DoxyPEP was made available to women who become pregnant, the risk of teratogenic effects is a theoretical possibility, especially if follow-up is infrequent. Ensuring that women of childbearing age are adequately monitored and informed about these risks is critical.

The cost of DoxyPEP also poses a considerable burden. Healthcare access varies globally, with different countries providing different levels of health insurance and universal healthcare coverage. The indirect costs associated with DoxyPEP can include additional health visits, screening and vaccinations, which might be prohibitive for some. The current DoxyPEP guidelines recommend two to four annual visits, which may result in individuals foregoing DoxyPEP. However, acts such as the Patient Protection and Affordable Care Act of 2010 in the US can overcome some of the costs.

Lastly, sexual health behaviors of individuals can evolve over time, a factor that must be considered when assessing the long-term efficacy and acceptability of DoxyPEP. While the ANRS-IPERGAY and ANRS DOXYVAC trials did not report any significant changes to sexual behavior, studies on HIV PrEP have documented behavioral shifts [63,64]. For instance, some individuals reported an increase in safe sex practices with the initiation of HIV PrEP, possibly due to increased awareness of their sexual health risks, whereas other individuals reported reduced condom use [63]. Furthermore, behaviors in these trials may not reflect real-life scenarios due to differences in context, monitoring and external influences.

## 11. The Future

When considering the future use of DoxyPEP, it is important to note that while it has proven effective in reducing the incidence of STIs like syphilis and chlamydia, its effectiveness against gonorrhea has been inconsistent [18]. This limitation highlights the necessity of combining DoxyPEP with additional preventative measures to improve overall public health outcomes.

One promising avenue for improving the effectiveness of STI prophylaxis involves combining DoxyPEP with vaccines. Research has shown the potential of meningococcal group B vaccines, developed for protection against *N. meningitidis*, to provide some level of cross-protection against *N. gonorrhoeae* due to shared antigens [65]. The 4CMenB vaccine has shown efficacy of up to 40% against *N. gonorrhoeae* when administered in two doses [66]. In the ANRS DOXYVAC study, Molina et al. studied the efficacy of both the 4CMenB vaccine and DoxyPEP, as DoxyPEP in the ANRS-IPERGAY substudy was found to be ineffective. Although the 4CMenB vaccine did not offer additional protection in this study [28], other studies (NCT04415424 and NCT04350138) in USA, Thailand and Australia are further researching this topic. Furthermore, additional research is looking at utilizing Generalized Modules for Membrane Antigens (GMMA)-based vaccines for the prevention of gonorrhea. A trial is currently underway (NCT05630859) where researchers are studying the safety and efficacy of a GMMA-based *N. gonorrhoeae* vaccine in several locations globally, including United States, Brasil, South Africa, Philippines and several sites within Europe.

Beyond vaccine integration, future considerations for DoxyPEP should include a broader approach targeted at diverse populations. Current studies have primarily focused on high-risk groups, and DoxyPEP was found to be effective in MSM populations [18,28], but not in cisgender women in Kenya [29]. These studies provide valuable insights; however, the need to further examine the effectiveness of DoxyPEP in populations other than MSM still remains, particularly as the study in Kenya found that adherence to DoxyPEP was low. This is of particular importance in sub-Saharan Africa, where individuals face a disproportionately high burden of STIs, and research must address how DoxyPEP could be tailored to meet the needs of these communities. Understanding how different populations experience STI risks and the potential benefits of DoxyPEP can help ensure that preventive strategies are both effective and equitable. Additionally, if low adherence presents a barrier, then additional education and, possibly, different prevention strategies may need to be identified.

Ongoing AMR surveillance is essential while administering DoxyPEP, particularly when it is deployed in controlled settings. Doxycycline is included in the WHO list of essential medicines and is indicated for the treatment of numerous infections, apart from chlamydia and syphilis [27]. Doxycycline resistance has been found in a number of clinically important species, including *S. aureus*, *M. genitalium*, *S. pneumoniae* and *Haemophilus influenzae* [67]. As DoxyPEP is rolled out to more countries as a preventative measure against STIs, increased monitoring for AMR is essential to safeguard its effectiveness in other clinical settings.

In conclusion, while DoxyPEP presents a promising approach to reducing the incidence of bacterial STIs among high-risk populations, its implementation must be carefully managed to avoid potential pitfalls. Comprehensive strategies that combine DoxyPEP with other preventive measures, such as vaccines, sexual health education, and screening, will be crucial to address the evolving landscape of STIs. Lastly, integration of DoxyPEP into the unique constraints and burdens of individual countries should be an important focal point of future research.

## Figures and Tables

**Table 1 antibiotics-13-01183-t001:** Summary of doxycycline post-exposure prophylaxis (DoxyPEP) studies for sexually transmitted infections.

Study	Population	Sample Size	Duration	Effectiveness of DoxyPEP *		
				**Chlamydia**	**Syphilis**	**Gonorrhea**
ANRS-IPERGAY (France): [18]	MSM and TGW who have sex with men without HIV on HIV PrEP	232	Medium follow-up: 8.7 months	HR: 0.30 (95% CI 0.13–0.70; *p* = 0.006)	HR: 0.27 (95% CI 0.007–0.98; *p* = 0.047)	HR: 0.83 (95% CI 0.47–1.47; *p* = 0.52)
Luetkemeyer et al. (US) [30]	MSM and TGW who have sex with men taking HIV PrEP ^1^, or PLWH ^2^	501	Medium follow-up: 9 months	RR: 0.12 (95% CI 0.05–0.25) ^1^	RR: 0.13 (95% CI 0.03–0.59) ^1^	RR: 0.45 (95% CI 0.32–0.65) ^1^
				RR: 0.26 (95% CI 0.12–0.57) ^2^	RR 0.23 (95% CI 0.04–1.29) ^2^	RR: 0.43 (95% CI 0.26–0.71) ^2^
Stewert et al. (Kenya) [29]	Cisgender women in Kenya on HIV PrEP	449	12 months	RR: 0.73 (95% CI 0.47–1.13)	–	RR: 1.64 (95% CI 0.78–3.47)
ANRS DOXYVAC (France) [28]	MSM and TGW who have sex with men without HIV on HIV PrEP	556	Medium follow-up: 14 months	aHR: 0.14 (95% CI 0.09–0.23; *p* < 0.0001)	aHR: 0.21 (95% CI 0.11–0.41; *p* < 0.0001)	aHR: 0.67 (95% CI 0.52–0.87; *p* = 0.0025)

* Efficacy against the first incidence of Chlamydia, syphilis or gonorrhea; MSM: Men who have sex with men; TGW: Transgender women; HR: Hazard ratio; aHR: Adjusted hazard ratio; ^1^ PrEP: Pre-exposure Prophylaxis; ^2^ PLWH: People living with HIV; RR: Relative risk.

## Data Availability

No new data were created or analyzed in this study.

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
