# Peer review of "From Cure to Prevention: Doxycycline’s Potential in Prophylaxis for Sexually Transmitted Infections"

_antibiotics, 2024, doi:10.3390/antibiotics13121183_

Round 1

Reviewer 1 Report

Comments and Suggestions for Authors

Dear Editor,

Thank you for the opportunity to revise the manuscript " From Cure to Prevention: Doxycycline’s Potential in Sexually Transmitted Infections’ Prophylaxis"

In this manuscript, the authors describe recent advances in the field of DoxyPEP. Other very recent systematic reviews have been published on this topic, which appear to be very similar to the outbreak presented.

I therefore have some comments for the authors which I hope will help to strengthen this manuscript.

Abstract. The risk of AMR among pathogens other than sexually transmitted diseases

Introduction.

Please refer to lines 35-36. Please provide more details and historical context.

Line 38 Please provide more details such as dosage and regimen.

Line 40 Please refer to data on 4CMenB and gonorrhoea when discussing vaccines.

Lines 43-44. The authors might acknowledge that tolerability is also a key factor in the choice of doxy. 

Line 60. The reduction in testing and screening is the most likely explanation other than changes in sexual behaviour. A global disruption in testing rather than transmission seems to be the case.

Lines 93-94. Doxy followed by Moxi is the first-line treatment for Mgen infection according to CDC guidelines in the absence of molecular AMR testing.

Line 95. Please include reference.

Line 96. Doxy is the second line of treatment for syphilis and is actually used very often (lack of penicillin due to global shortages, allergy to penicillin, etc).

Line 97. This is not suggested by "some studies" but is actually part of the STI guidelines.

Line 103.  To avoid confusion, the authors may refer to 'condomless' rather than 'unprotected', as among PrEP users, for example, 'protect sex' may refer to the use of PrEP itself. 

Line 124. Please rephrase "attrition" for clarity.

Line 129. This is a substudy of the ANRS_IPERGAY study. The authors could clarify this. 

Lines 149-154. The authors could also include the 4CMenB results when discussing the DOXYVAC trial.

Table 1. The authors could consider including efficacy data by different STIs (Tp, Ct, Ng) in the table.

Line 163. The authors mention lack of testing services as one explanation for the DoxyPEP results in women in Africa. However, as part of the RCT, women were indeed tested at planned visits, which is not related to existing services for the general population. 

Lines 167-169. There is evidence of doxy concentrations at the vaginal site presented at recent international congresses, please consider providing a reference. 

Lines 180-182. Please acknowledge the number of culture-positive gonorrhoea samples in this study and recognise the small number as a limitation.

Lines 192-194. Please cite the recently published work assessing the potential development of resistance in vitro (which was not observed) when Treponema pallidum was exposed to suboptimal levels of doxycicline (also for Lines 345-346)

Lines 196-197. There is no evidence of this, even in the general aprt of DoxyPEP. 

Lines 210-212. However, doxy is not a first-line treatment for pneumococcal infections.

When discussing resistance concerns in the area of STIs, please consider adding the following:

Chlamydia: 1) Resistance to doxycycline has never been documented in clinical practice. 2) However, doxy-R has been seen in C. suis, which could potentially be transferred to other chlamydia species. 

Syphilis: 1) Resistance to doxycycline has never been documented in clinical practice.  2) Possible TCN point mutations. 3) Possible effect on syphilis serologies by delaying syphilis seroconversion when using DoxyPEP. 

Mgen: Resistance to doxycycline has never been documented in clinical practice.

Lines 226-227. However, no clinical evidence of this is reported.

Lines 231-246. Make it clear that this concept is theoretical, with no real-world evidence. 

The authors might also consider briefly discussing the real need for screening, treatment and thus prevention of asymptomatic chlamydia and gonorrhoea in MSM (see GONOSCREEN study results and related literature).

The BASHH guidelines (or UK Racommendation) do not appear to have been updated.

Please include the Australian, IUSTI and EACS guidelines on DoxyPEP.

As several reviews on DoxyPEP have been published, please include more data on the real-world use of DoxyPEP. This seems to be crucial to demonstrate the novelty of the manuscript. 

The authors kindly acknowledge the results presented at CROI 2024, Denver. However, please consider presenting data on real life use of DoxyPEP presented at recent congresses such as IAS congress, Munich and HIV R4P, Lima, Peru. 

Line 376. There is uncertainty about the teratogenic risk of doxy. The authors may wish to rephrase "significant". 

Line 382. Doxy is very cheap in most settings. Please consider indirect costs such as visits, STI screening, PrEP and vaccines. 

Lines 387-389. However, acknowledge that real life may be different from these RCTs. 

Reference number 52 appears to be out of context, please consider using a reference to 4CMenB and gonorrhoea. 

Line 408. Please include reference to ongoing trials of GMMA vaccines on Ng.

Lines 413. One RCT showed no efficacy in women. It may be that we need a different prevention strategy than DoyPEP for cisgender African women. 

Author Response

Dear Reviewer

Thank you for taking your time to thoroughly review our manuscript. Please find the detailed responses below and the corresponding revisions and corrections in the re-submitted files. For each change, the line number has been recorded in the response.

Comment 1:      Abstract. The risk of AMR among pathogens other than sexually transmitted diseases

Response 1:       Thank you, we agree and have included that AMR is a risk in other infections as well, including K. pneumoniae and S. aureus. It now reads: “Key issues involve the potential development of antimicrobial resistance, particularly among common STIs pathogens like C. trachomatis, M. genitalium, and N. gonorrhoeae, as well as other bacteria such as S. aureus and K. pneumoniae.”

Comment 2:      Please refer to lines 35-36. Please provide more details and historical context.

Response 2:      Thank you for your comment. Additional details have been added. Particularly, we added the antibiotic regimen. Lines 36-37: “These antibiotic regimens often consisted of azithromycin with or without ciprofloxacin and cefixime” and types of STIs included “These antibiotic regimens often consisted of azithromycin with or without ciprofloxacin and cefixime”. However, please note that additional data such as the countries or dates in which the studies were done were not mentioned in the original paper.

Comment 3:      Line 38 Please provide more details such as dosage and regimen.

Response 3:      Thank you for your comment. Additional details have been added with regards to the results of the strategy. Line 41: “and resulted in rapid selection and dissemination of antibiotic resistance to sulphonamides, minocycline, and penicillin.” ; however, please note that the dosages and regimen were not mentioned in the original paper.   

Comment 4:      Line 40 Please refer to data on 4CMenB and gonorrhoea when discussing vaccines.

Response 4:      Thank you for your comment. We have added an additional part briefly discussing the role of vaccines and mentioned the 4CMenB vaccine as per your feedback. Line 43: “Despite the success of 4CMenB vaccine against Neisseria meningitides, showing 71%-95% vaccine effectiveness in infants [55], a similar success has not been seen in Neisseria gonorrhoea. Additionally, there no vaccines against chlamydia and syphilis, among other STIs.”

Comment 5:      Lines 43-44. The authors might acknowledge that tolerability is also a key factor in the choice of doxy. 

Response 5:      Thank you for your comment. We do mention the tolerability of doxycycline in the adverse effects section, but we have added “tolerability” (line 51) as per your feedback in the introduction to be more comprehensive.

Comment 6:      Line 60. The reduction in testing and screening is the most likely explanation other than changes in sexual behaviour. A global disruption in testing rather than transmission seems to be the case.

Response 6:      Thank you for pointing this out. I have edited this, and further added a reference supporting the statement. Line 61: “The US Centers for Disease Control and Prevention (CDC) has observed a consistent year-on-year increase in the rates of these STIs. However, there was a notable exception from 2020 to 2021, in which the implementation of lockdowns and social distancing measures due to the COVID-19 pandemic led to a temporary decrease in STI prevalence rates [7 ]. This pattern was similarly observed by the European Centre for Disease Prevention and Control (ECDC) [8– 10], indicating a global disruption in STI screening during the pandemic [11].”

Comment 7:      Lines 93-94. Doxy followed by Moxi is the first-line treatment for Mgen infection according to CDC guidelines in the absence of molecular AMR testing.

Response 7:      Thank you for your comment. This has been rectified. Lines 97-98: “According to CDC guidelines, Doxycycline is the first-line treatment for M. genitalium infections, along with moxifloxacin or azithromycin, depending on the sensitivity, and for C. trachomatis infections.

Comment 8:      Line 95. Please include reference.

Response 8:      Thank you for your comment. We have added a reference to the statement. Line 99: “Workowski, K.A.; Bachmann, L.H.; Chan, P.A.; Johnston, C.M.; Muzny, C.A.; Park, I.; Reno, H.; Zenilman, J.M.; Bolan, G.A. Sexually Transmitted Infections Treatment Guidelines, 2021. MMWR Recomm Rep 2021, 70. https://doi.org/10.15585/mmwr.rr7 568004a1.”

Comment 9:      Line 96. Doxy is the second line of treatment for syphilis and is actually used very often (lack of penicillin due to global shortages, allergy to penicillin, etc).

Response 9:      Thank you for your comment. This has been rectified. Lines 100-102: “ With regards to syphilis treatment, doxycycline is second line treatment, often used in cases of penicillin allergy or shortages”

Comment 10:        Line 97. This is not suggested by "some studies" but is actually part of the STI guidelines.

Response 10:       Thank you for your comment. This has been rectified with the comment from above.

Comment 11:       Line 103.  To avoid confusion, the authors may refer to 'condomless' rather than 'unprotected', as among PrEP users, for example, 'protect sex' may refer to the use of PrEP itself. 

Response 11:        Thank you for your comment. We have replaced “unprotected” (Line 108) with “condomless” as per your feedback.

Comment 12:        Line 124. Please rephrase "attrition" for clarity.

Comment 12:        I have rephrased for clarity. Changed to “drop-out rates” (Line 129).

Comment 13:        Line 129. This is a substudy of the ANRS_IPERGAY study. The authors could clarify this. 

Comment 13:       Thank you for your comment. We have clarified this as per your feedback. Line 134: “A stubstudy of The French ANRS-IPERGAY trial conducted in 2015 assessed the efficacy of DoxyPEP in HIV-negative MSM and TGW who engage in sex with men”

Comment 14:      Lines 149-154. The authors could also include the 4CMenB results when discussing the DOXYVAC trial.

Response 14:       Thank you for your comment. The 4CMenB results in the DOXYVAC trial were discussed in the section “the future”; lines 489-493. We mentioned it there as we were discussing the role of vaccines as a future prophylaxis strategy.

Comment 15:       Table 1. The authors could consider including efficacy data by different STIs (Tp, Ct, Ng) in the table.

Response 15:        Thank you for your comment. We have added efficacy data by different STI to the table for each study as per your feedback.

Comment 16:        Line 163. The authors mention lack of testing services as one explanation for the DoxyPEP results in women in Africa. However, as part of the RCT, women were indeed tested at planned visits, which is not related to existing services for the general population. 

Response 16:        Thank you for your comment. In line 163, with regards to the testing services, we were referring to women’s partners and not the subjects of the study themselves. “Lack of services for testing and treating partners”

Comment 17:        Lines 167-169. There is evidence of doxy concentrations at the vaginal site presented at recent international congresses, please consider providing a reference. 

Response 17:        Thank you for your comment. This part has been re-written for clarification and consistency with the literature and a reference has been added as well. Lines 171-178: “Moreover, it has been proposed that differences in primary infection site may contribute to the differences in DoxyPEP efficacy between cisgender women and MWS and TGW. In cisgender women, the primary site of infection is the endocervices; while, in MSM and TGW the primary site of infection are rectal tissue and pharyngeal tissue. However, recent evidence has suggested that doxycycline concentrations in the vaginal tissue were sufficient enough to prevent STIs. This finding suggests that another factor other than differences in site of infection may contribute to the difference in efficacy; nonetheless, this would need further investigation”

Comment 18:        Lines 180-182. Please acknowledge the number of culture-positive gonorrhoea samples in this study and recognise the small number as a limitation.

Response 18:        Thank you for your response. We have added that number of isolates as per your feedback. Lines 204-205: “however, its important to note that only 78 isolates were available, serving as one of the limitation of this study”

Comment 19:       Lines 192-194. Please cite the recently published work assessing the potential development of resistance in vitro (which was not observed) when Treponema pallidum was exposed to suboptimal levels of doxycicline (also for Lines 345-346)

Response 19:       Thank you for your comment. To clarify that doxycycline exhibits antimicrobial activity against syphilis we have added. Lines 216-217: “Nonetheless, in vitro studies have shown that doxycycline exhibits anti-treponemal activity (MIC 0.10 mg/L)”.

Comment 20:       Lines 196-197. There is no evidence of this, even in the general aprt of DoxyPEP. 

Response 20:       Thank you for your comment. We have removed line 196 to better reflect the current evidence. Lines 217-220: “In summary, currently only N. gonorrhoea has developed tetracycline resistance, which may be further exacerbated. Nonetheless, further studies with longer follow-ups are needed to accurately determine the risk of AMR with enrolment of DoxyPEP”

Comment 21:       Lines 210-212. However, doxy is not a first-line treatment for pneumococcal infections.

Response 21:       Thank you for your comment. While doxy indeed is not a first line treatment for pneumococcal infections, we decided to mention pneumococci to highlight the how far the effects of DoxyPEP could potentially extend.  In addition, it is often used for community acquired respiratory tract infections, including those caused by pneumococci (Line 228).

Comment 22:        When discussing resistance concerns in the area of STIs, please consider adding the following:

Chlamydia: 1) Resistance to doxycycline has never been documented in clinical practice. 2) However, doxy-R has been seen in C. suis, which could potentially be transferred to other chlamydia species. 

Syphilis: 1) Resistance to doxycycline has never been documented in clinical practice.  2) Possible TCN point mutations. 3) Possible effect on syphilis serologies by delaying syphilis seroconversion when using DoxyPEP. 

Mgen: Resistance to doxycycline has never been documented in clinical practice.

Response 22:       Thank you for your comment. We have added the following to the antibiotic resistance section as per your feedback. Line 179 “Nevertheless, currently, there have been no documented clinical cases of doxycycline resistant chlamydia, syphilis or mycoplasma genitalium”. Lines 255-259 “As previously mentioned, there have been no documented clinical cases of doxycycline resistant chlamydia. However, a study investigated C. suis isolated from pigs for tetracycline resistant genes, and found that tet(C) was present in all isolates. The exact mechanism in which these isolates acquired resistance is unknown. Nonetheless, this finding reflects the theoretical possibility that chlamydia trachomatis could acquire tetracycline resistance.” And Lines 259-266: “Unlike other bacteria, T. pallidum does not acquire resistance through gene transfer, but rather via point mutations. This has already been observed within the 23s rRNA gene resulting in azithromycin resistance. In other organisms, a similar mutation in the 16s rRNA gene results in doxycycline resistance. Moreover, studies have shown that odds of T. pallidum resistance are higher in individuals who have taken macrolide therapy in the preceding year. Accordingly, concerns remain about the emergence of such mutations with the widespread use of Doxy-PEP”.

Comment 23:        Lines 226-227. However, no clinical evidence of this is reported.

Response: 23:       Thank you for your comment. We were discussing potential problems that may arise with DoxyPEP in regard to AMR, and not current issues. Nevertheless, we have further clarified that and emphasised that lack of clinical evidence for that point. Lines 253-254: “Nonetheless, no clinical evidence of this has been reported”

Comment 24:       Lines 231-246. Make it clear that this concept is theoretical, with no real-world evidence. 

Response 24:       Thank you for your comment. We have clarified that this is a hypothetical concept. Lines 271-272: “This hypothetical concept is known as the "Arrested Immunity Hypothesis" and was initially based on the interaction between the immune system and chlamydia”

Comment 25:      The authors might also consider briefly discussing the real need for screening, treatment and thus prevention of asymptomatic chlamydia and gonorrhoea in MSM (see GONOSCREEN study results and related literature).

Response 25:       We agree. We have included a short paragraph regarding screening, including a brief discussion of the GONOSCREEN study, within the “Ethical Considerations” section: Lines 437-446: “STI screening is a vital part of preventative healthcare, which is highlighted in many of the guidelines recommending DoxyPEP. The Belgium GONOSCREEN study compared the effect of screening for gonorrhoea and chlamydia to non-screening [59 ]. The study demonstrated that non-screening was associated with a higher incidence of asymptomatic chlamydia infections, emphasizing the role of regular screening in identifying and treating these cases before they contribute to ongoing transmission. However, it also highlighted that screening did not significantly reduce gonorrhoea incidence. Notably, this study was associated with increased antimicrobial consumption, raising ethical concerns regarding the overprescription of antibiotics, mirroring reservations surrounding the wider use of DoxyPEP.”

Comment 26:       The BASHH guidelines (or UK Racommendation) do not appear to have been updated.

Response 26:       Thank you, I have rephrased the sentence to better clarify that the guidelines have not been updated since 2021. Lines 285-287: “The guidance from the British Association for Sexual Health and HIV, last updated in 2021, advises against routinely prescribing doxycycline as prophylaxis for STIs”

Comment 27:      Please include the Australian, IUSTI and EACS guidelines on DoxyPEP.

Response 27:      Thank you, this will better outline the ongoing consensus between organizations in regard to DoxyPEP. we have included the guidelines from the three organizations (Lines 308-317, 332-349).

Comment 28:       As several reviews on DoxyPEP have been published, please include more data on the real-world use of DoxyPEP. This seems to be crucial to demonstrate the novelty of the manuscript. 

Response 28:       Thank you for your comment. We have done extensive research throughout the literature but have failed to find more examples of real-world use of DoxyPEP in the STI setting. The only example we found was the San Francisco DoxyPEP implementation presented at CORI 2024, which we have already included in our paper. 

Comment 29:       The authors kindly acknowledge the results presented at CROI 2024, Denver. However, please consider presenting data on real life use of DoxyPEP presented at recent congresses such as IAS congress, Munich and HIV R4P, Lima, Peru. 

Response 29:       Thank you for your comment. We have added data presented in the IAS congress 2024. The two trails that were presented at the conference (one from Canada and one from Japan) focused on Doxy-PrEP, and not DoxyPEP. However, we did find a qualitative study about DoxyPEP adherence which we added to the paper. “ A qualitative study assessing DoxyPEP adherence in women in western Kenya presented its findings in the International AIDS society (IAS) conference 2024. It revealed that the main barriers to DoxyPEP adherence were side effects, mainly nausea, misunderstanding dosage instructions, and fear of partner reaction discouraged proper adherence. On the other hand, the study revealed that the perceived value of preventing an STI, familiarity with doxycycline, and the use of a discrete pill case motivated some women to adhere to DoxyPEP. These findings highlight the complexity of implementing DoxyPEP in some parts of the world, emphasizing the importance of ensuring patient understanding, being culturally competent, and battling stigma.”

Comment 30:       Line 376. There is uncertainty about the teratogenic risk of doxy. The authors may wish to rephrase "significant". 

Response 30:       We have rephrased it to state that the risk is a theoretical possibility.  Lines 458-461: “If DoxyPEP were made available to women who become pregnant, the risk of teratogenic effects is a theoretical possibility, especially if follow-up is infrequent.”

Comment 31:        Line 382. Doxy is very cheap in most settings. Please consider indirect costs such as visits, STI screening, PrEP and vaccines. 

Response 31:        Thank you, we have rephrased it to better emphasize that the indirect costs may be burdensome. Lines 464-465: “The indirect costs associated with DoxyPEP can include additional health visits, screening and vaccinations, which might be prohibitive for some.”

Comment 32:         Lines 387-389. However, acknowledge that real life may be different from these RCTs. 

Response 32:         Thank you. We have included a statement to further explain the possible differences in behaviours between real life and trials.  Lines 475-477: “Furthermore, behaviours in these trials may not reflect real-life scenarios due to differences in context, monitoring and external influences.”

Comment 33:        Reference number 52 appears to be out of context, please consider using a reference to 4CMenB and gonorrhoea. 

Response 33:        Thank you. We have changed the reference to a more suitable one, and adjusted the phrasing. The reference is now. Line 487: “Petousis-Harris, H., Paynter, J., Morgan, J., Saxton, P., McArdle, B., Goodyear-Smith, F., Black, S., 2017. Effectiveness of a group B outer membrane vesicle meningococcal vaccine against gonorrhoea in New Zealand: a retrospective case-control study. The Lancet 390, 1603–1610. https://doi.org/10.1016/S0140-6736(17)31449-6“

Comment 34:       Line 408. Please include reference to ongoing trials of GMMA vaccines on Ng.

Response 34:       Thank you, We have included a reference and a small statement on this important trial (NCT05630859 – Line 495).

Comment 35:        Lines 413. One RCT showed no efficacy in women. It may be that we need a different prevention strategy than DoyPEP for cisgender African women. 

Response 35:        Thank you. We have rephrased the paragraph to better capture the importance of adherence, and different prevention strategies that may be required in different populations. Lines 501-504: “These studies provide valuable insights, however the need to further examine the effectiveness of DoxyPEP in populations other than MSM still remain, particularly as the study in Kenya found that adherence to DoxyPEP was low.”

We have also included the statement: Lines 508-510: “Additionally, if low adherence presents as a barrier, then additional education, and possibly, different prevention strategies may need to be identified”

Reviewer 2 Report

Comments and Suggestions for Authors

I would like to thank for the opportunity to review this interesting manuscript. J Bird et al. provide an overview of the research evaluating doxycycline as post-exposure prophylaxis for prevention sexually transmitted infections and address concerns regarding doxycycline safety, emergence of resistance, implementation and relevant ethical considerations. 

              The authors provide a well-rounded and clinically relevant review of the evidence. Although not the focus of the paper, a brief discussion of the use of doxycycline as preexposure STI prophylaxis would be pertinent

Some points that should be addressed: 

1.        Line 31. A more appropriate reference regarding the failure of the test and treat strategy should be added

2.        Line 37. A more detailed review of examples of past use of STI prophylaxis is relevant. Issues encountered in the past may be challenges we face today

3.        Lines 48-50. The phrase “It aims…prevention.”  seems redundant.

4.        Line 54. “were particularly concerning” is repeated many times. Should be removed

5.        Line 55. “in the previous year”. It creates confusion as it is stated in the that this data come from 2021

6.        Line 57. Rather more specifically than “additionally”

7.        Line 64. A reference should be added

8.        Line 82. A more recent reference should be available

9.        Line 93. Please add the guideline the recommendations stem from

10.  Line 96. Penicillin shortages have resulted in substantial increase in experience with doxycycline for treatment of syphilis.

11.  Line 147. The explanation regarding the observed lack of efficacy in gonorrhea contradicts the positive findings of DOXYVAC

12.  Line 167. The statement needs elaboration. Other limitations apply such as the level of doxy resistance in gonorrhea in Kenya

13.  Line 169. Meta-analyses of RCT data with regard to the efficacy of doxycycline as STI prophylaxis have been published and should be included. Also it would seem pertinent to add here data on the effectiveness of the intervention which currently are described in the implementation challenges section.

14.  Line 176. Also selection of cross-resistance in gonorrhea is a major concern

15.  Line 218. Not similar MICs  but a positive association

16.  Line 349. Some ethical considerations coincide with implementation challenges. Perhaps the two could be discussed in the same section

17.  Line 405. DoxyPEP did protect against gonorrhea in the context of DoxyVAC study. 

Author Response

Dear Reviewer

Thank you for taking your time to thoroughly review our manuscript. Please find the detailed responses below and the corresponding revisions and corrections in the re-submitted files. For each change, the line number has been recorded in the response.

Comment 1:     Line 31. A more appropriate reference regarding the failure of the test and treat strategy should be added

Response 1:      Thank you for your comment. A new, more appropriate reference has been added in its place. Line 31: “Nah K, Nishiura H, Tsuchiya N, Sun X, Asai Y, Imamura A. Test-and-treat approach to HIV/AIDS: a primer for mathematical modeling. Theor Biol Med Model. 2017;14(1):16. Published 2017 Sep 5. doi:10.1186/s12976-017-0062-9”

Comment 2:       Line 37. A more detailed review of examples of past use of STI prophylaxis is relevant. Issues encountered in the past may be challenges we face today

Response 2:       Thank you for your comment. Reviewer 1 had the same comments and we have added the following details: Antibiotic regimens in Line 36-38: “These antibiotic regimens often consisted of azithromycin with or without ciprofloxacin and cefixime”, and types of bacteria involved “gonorrhoea and chlamydia, but not of syphilis” and for the US Navy use of STI prophylaxis we added the following “and resulted in rapid selection and dissemination of antibiotic resistance to sulphonamides, minocycline, and penicillin”.

With regards to past challenges that we may face today, these are discussed in the section “Challenges to widespread implementation”.

Comment 3:        Lines 48-50. The phrase “It aims…prevention.”  seems redundant.

Response 3:        Thank you for your comment. We have removed this line to avoid redundancy.

Comment 4:        Line 54. “were particularly concerning” is repeated many times. Should be removed

Response 4:        Thank you. We have removed “were particularly concerning” and rephrased the sentence. It now states: Lines 58-60: “. According to the World Health Organization (WHO), in 2020, STIs such as gonorrhea, chlamydia, and syphilis accounted for more than 374 million cases worldwide [6], underscoring their widespread impact on global health.”

Comment 5:        Line 55. “in the previous year”. It creates confusion as it is stated in the that this data come from 2021

Response 5:        We agree. We have changed the date, and adjusted the sentence to better confer the message across. Lines 58-60: “According to the World Health Organization (WHO), in 2020, STIs such as gonorrhea, chlamydia, and syphilis accounted for more than 374 million cases worldwide, underscoring their widespread impact on global health.”

Comment 6:        Line 57. Rather more specifically than “additionally”

Response 6:        Thank you, I Have removed the word “additionally”. It now reads: Lines 61-62: “The US Centers for Disease Control and Prevention (CDC) has observed a consistent year-on-year increase in the rates of these STIs”

Comment 7:        Line 64. A reference should be added

Response 7:        I have included a reference, thank you. Line 69: “United States Center for Disease Control and Prevention. Sexually Transmitted Infections Surveillance 2022. Technical report, US Department of Health and Human Services, Atlanta, 2024.”

Comment 8:        Line 82. A more recent reference should be available

Response 8:        I have updated the reference to a more recent one. Line 86: “Moreno de Lara, L.; Parthasarathy, R.S.; Rodriguez-Garcia, M. Mucosal Immunity and HIV Acquisition in Women. Current Opinion in Physiology 2021, 19, 32–38. https://doi.org/10.1016/j.cophys.2020.07.021.”

Comment 9:        Line 93. Please add the guideline the recommendations stem from

Response 9:       Thank you for your comment. We have rectified this point, the guidelines referenced here are from the CDC, with the reference: Line 99: Workowski, K.A., Bachmann, L.H., Chan, P.A., Johnston, C.M., Muzny, C.A., Park, I., Reno, H., Zenilman, J.M., Bolan, G.A., 2021. Sexually Transmitted Infections Treatment Guidelines, 2021. MMWR Recomm Rep 70. https://doi.org/10.15585/mmwr.rr7004a1

Comment 10:        Line 96. Penicillin shortages have resulted in substantial increase in experience with doxycycline for treatment of syphilis.

Response 10:        Thank you for your comment. We have added this point. Lines 100-102: “Doxycycline is second line treatment, often used in cases of penicillin allergy or shortages, particularly in low- and middle-income countries.”

Comment 11:          Line 147. The explanation regarding the observed lack of efficacy in gonorrhoea contradicts the positive findings of DOXYVAC

Response 11:          Thank you for your comment. We have re-written to ensure clarity and consistency. Lines 155-161“the efficacy of DoxyPEP in MSM on HIV PrEP and revealed that DoxyPEP was associated with a significant reduction in the incidence of chlamydia (aHR: 0.13 [95% CI 0.08-0.21]; p<0.0001) and syphilis (aHR: 0.22 [95% CI 0.11-0.43]; p<0.0001). On the other hand, the study showed moderate, but lower, efficacy against gonorrhea (aHR: 0.63 [95% CI 0.50-0.79]; p<0001)(Table 1) when compared to the US DoxyPEP study [30]. This further reflects the effect of tetracycline resistance on the efficacy of DoxyPEP against gonorrhea.”

Comment 12:            Line 167. The statement needs elaboration. Other limitations apply such as the level of doxy resistance in gonorrhoea in Kenya

Response 12:            Thank you for your comment. We have added this point. Line 169: “and high prevalence of tetracycline resistant Neisseria gonorrhoea”.

Comment 13:            Line 169. Meta-analyses of RCT data with regard to the efficacy of doxycycline as STI prophylaxis have been published and should be included. Also it would seem pertinent to add here data on the effectiveness of the intervention which currently are described in the implementation challenges section.

Response 13:            Thank you for your comment. We have added a meta-analysis of the studies included as per your feedback. Lines 179-186: “A meta-analysis by Szondy et al. assessed the current state of evidence on the efficacy of DoxyPEP. Among other studies, the meta-analysis included the ANRS-IPERGAY trail [17 ], the US DoxyPEP trial [29 ], the DoxyPEP Kenya Trial [28 ], and the DOXYVAC trial [27]. The results of the analysis remained consistent with the findings in each of the studies, with an increased incidence of STI in the control group (RR: 0.47 [95% CI 0.29-0.77]). More specifically, the incidence of chlamydia (RR: 0.26 [95% CI 0.10-0.68]), Syphilis (RR: 0.23 [95% CI 0.14-0.36]), and gonorrhoea (RR: 0.66 [95% CI 0.34-1.26]) were all lower in the DoxyPEP group when compared to the control group [30]”

Comment 14:              Line 176. Also selection of cross-resistance in gonorrhea is a major concern

Response 14:               Thank you for your comment. Cross resistance in gonorrhoea is discussed in detail in the same section, lines 235-245. However, we did add “With doxycycline resistant gonorrhea already prevalent, DoxyPEP can further exacerbate this issue and increase the risk of cross-resistance in gonorrhea” as part of the introduction to this section. 

Comment 15:              Line 218. Not similar MICs  but a positive association

Response 15:               Thank you for your comment. This has been rectified as per your feedback. Line 240: “ was found to have a MICs with a positive association”

Comment 16:               Line 349. Some ethical considerations coincide with implementation challenges. Perhaps the two could be discussed in the same section

Response 16:               Thank you for your comment. We did debate this and felt that there were sufficient differences to keep the two sections.

Comment 17:                Line 405. DoxyPEP did protect against gonorrhea in the context of DoxyVAC study. 

Response 17:                Thank you, we have rephrased the section to better clarify that DoxyPEP was found to be ineffective in the ANRS-IPERGAY study. Lines 484-497: “In the ANRS DOXYVAC study, Molina et al. (2024) studied the efficacy of both the 4CMenB vaccine and DoxyPEP, as DoxyPEP in the ANRS-IPERGAY substudy was found to be ineffective. Although the 4CMenB vaccine did not offer additional protection in this study, other studies (NCT04415424 and NCT04350138) in USA, Thailand and Australia are further researching this topic

Round 2

Reviewer 1 Report

Comments and Suggestions for Authors

Thank you for addressing my comments.